# Ipragliflozin Ameliorates Diabetic Nephropathy Associated with Perirenal Adipose Expansion in Mice

**DOI:** 10.3390/ijms22147329

**Published:** 2021-07-08

**Authors:** Hideyuki Okuma, Kentaro Mori, Suguru Nakamura, Tetsuo Sekine, Yoshihiro Ogawa, Kyoichiro Tsuchiya

**Affiliations:** 1Interdisciplinary Graduate School of Medicine and Engineering, University of Yamanashi, Chuo 4093898, Japan; hokuma@yamanashi.ac.jp (H.O.); kmori@wustl.edu (K.M.); sugusamasaikou@yahoo.co.jp (S.N.); tsekine@yamanashi.ac.jp (T.S.); 2Department of Medicine and Bioregulatory Science, Graduate School of Medical Sciences, Kyushu University, Fukuoka 8168580, Japan; ogawa.yoshihiro.828@m.kyushu-u.ac.jp; 3Department of Diabetes and Endocrinology, University of Yamanashi Hospital, Chuo 4093898, Japan

**Keywords:** SGLT2 inhibitors, adipose tissue, diabetic nephropathy

## Abstract

Sodium glucose cotransporter-2 (SGLT2) inhibitors inhibit the development of diabetic nephropathy (DN). We determined whether changes in perirenal fat (PRAT) by a SGLT2 inhibitor ipragliflozin (Ipra) contribute to the suppression of DN development. High-fat diet (HFD)-fed mice were used as a DN model and were treated with or without Ipra for 6 weeks. Ipra treatment reduced urinary albumin excretion (UAE) and glomerular hypertrophy in HFD-fed mice. In the PRAT of Ipra-treated mice, adipocyte size was increased, and inflammation, fibrosis, and adipocyte death were suppressed. In conditioned medium made from PRAT (PRAT-CM) of Ipra-treated mice, the concentration of leptin was significantly lower than PRAT-CM of mice without Ipra treatment. Serum leptin concentration in renal vein positively correlated with UAE. PRAT-CM from HFD-fed mice showed greater cell proliferation signaling in mouse glomerular endothelial cells (GECs) than PRAT-CM from standard diet-fed mice via p38MAPK and leptin-dependent pathways, whose effects were significantly attenuated in PRAT-CM from Ipra-treated mice. These findings suggest that Ipra-induced PRAT expansion may play an important role in the improvement of DN in HFD-fed mice. In vitro experiments suggest that reduced PRAT-derived leptin by Ipra could inhibit GECs proliferation, possibly contributing to the suppression of DN development.

## 1. Introduction

Chronic kidney disease (CKD) is a common disease and is a risk factor for cardiovascular, cerebrovascular, and end-stage renal diseases (ESRD). Metabolic diseases, such as diabetes mellitus, hypertension, obesity, and dyslipidemia, have been strongly associated with CKD development, with diabetic nephropathy (DN) being one of the most prevalent primary diseases in ESRD [1].

Sodium glucose cotransporter 2 (SGLT2) inhibitors are oral hypoglycemic agents that promote urinary glucose excretion by inhibiting glucose absorption at the proximal tubule. They have rapidly gained popularity because of their insulin-independent glycemic control activities and the ability to induce caloric loss and increase insulin sensitivity. Large-scale clinical trials among patients with type 2 diabetes revealed that SGLT2 inhibitors prevented macrovascular events such as myocardial infarction, stroke, and heart failure [2,3] and also prevented diabetic kidney diseases such as DN [4]. The renoprotective effects of SGLT2 inhibitors in humans can be attributed to tubuloglomerular feedback and hyperfiltration [5], and inhibition of glomerular hyperfiltration [6]. In mice, oxidative stress [7] and ectopic fat accumulation in the kidney [8] were reportedly involved in the development of DN. Nevertheless, the mechanisms of these effects remain matters of debate.

The amount of adipose tissue around the heart, blood vessels, and kidneys is correlated with the risk of developing cardiovascular diseases [9]. Furthermore, the quantity of perirenal adipose tissue (PRAT) located between the renal fascia and renal capsule positively correlated with urinary albumin excretion (UAE) in obese rats [10] and negatively correlated with glomerular filtration rate in hypertensive humans [11]. PRAT reportedly activates the renin-angiotensin system in hypertensive and obese rabbits by applying physical pressure to the surrounding blood vessels, worsening obesity-related metabolic disorders such as hypertension, insulin resistance, and atherosclerosis [12]. In addition to the amount of PRAT, functional changes of PRAT including inflammation, and adipokines might help mediate DN; the amelioration of PRAT inflammation through genetic or exogenous inhibition of plasminogen activator inhibitor-1 (PAI-1) expression improved DN in high-fat diet (HFD)-fed mice [13]; an angiotensin II receptor blocker telmisartan inhibited leptin secretion from PRAT to ameliorate DN, accompanied with suppression of proliferative signaling in the kidney [14]. Other researchers, including our group, previously reported that SGLT2 inhibitors reportedly enhanced appetite and promoted epididymal fat (Epi) accumulation without deteriorating inflammation or fibrosis in mice [15,16,17], referred to as “healthy adipose expansion”. In terms of interactions between adipose tissue with other organs, we further showed that Ipra gave rise to healthy adipose expansion in mice perivascular adipose tissue while implanting adipose tissue with healthy adipose expansion into the perivascular area of an atherosclerosis model suppresses vascular remodeling [18]. We also reported that Ipra increased fat accumulation in Epi, but decreased fat accumulation in liver [17]. Thus, it has been suggested that changes in the quality and quantity of fat within or surrounding each organ may underlie the mechanism of organ protection by SGLT2 inhibitors. Nevertheless, it has not been determined whether SGLT2 inhibitors induce healthy adipose expansion in PRAT or affect the pathogenesis of DN by altering secretory factors from PRAT.

The aim of this study was to investigate whether SGLT2 inhibitors induce “healthy adipose expansion” in PRAT and whether it affects the pathogenesis of DN by altering PRAT-derived secretory factors.

## 2. Results

### 2.1. Ipra Increased Adipocyte Size While Attenuating Insulin Resistance in PRAT of HFD-Fed Mice

HFD-fed wild-type (WT) mice were divided into two groups, one treated with Ipra for 6 weeks and the other not. Similar to our previous findings [17,18], Ipra treatment did not affect body weight (Figure 1a), but ameliorated HFD-induced hyperglycemia (Figure 1b) accompanied by increased urinary glucose excretion (Figure 1c). Ipra treatment suppressed the increase of UAE in HFD-fed mice (Figure 1d), decreased serum insulin levels (Figure 1e), and improved insulin resistance (Figure 1f). Ipra increased serum non-esterified fatty acids and 3-hydroxybutyrate (3-HBA), suggesting increased lipolysis (Appendix A), and significantly increased the weight of Epi and decreased the weight of liver (Appendix A). Ipra significantly increased the weight of PRAT (Figure 1g). Weights of PRAT and Epi showed a strong inverse correlation with liver weight (Appendix A). Histological analysis showed that Ipra induced an increase in the size of adipocytes of PRAT in HFD-fed mice (Figure 1h,i), although there was no significant difference in the number of PRAT adipocytes between HFD-fed mice treated with and without Ipra (Figure 1j). Gene expression analysis of PRAT revealed that Ipra treatment significantly or tended to increase the expression of lipolysis- and lipogenesis-related genes, including *Atgl*, *Hsl*, and *Pck1* (Figure 1k).

As previously reported [15], IL-15 gene expression in PRAT was suppressed in Ipra-treated mice (Figure 1k). HFD feeding decreased expression of Perilipin-1, a marker of lipolysis, in the PRAT, although this decrease was partly negated by Ipra treatment, suggesting that Ipra treatment increased the lipolysis of PRAT (Figure 1l). Insulin-induced Akt phosphorylation of PRAT was decreased by HFD feeding, which was partly ameliorated by Ipra administration (Figure 1m). Taken together, these results indicate that Ipra reduces UAE as well as hyperglycemia in HFD-fed mice, and that Ipra increases lipogenesis and lipolysis capacity while enhancing insulin signaling in PRAT of HFD-fed mice.

### 2.2. Ipra Suppressed Inflammation, Fibrosis, and Cell Death in PRAT of HFD-Fed Mice and Shifted Macrophage Properties toward an Anti-Inflammatory State

We next examined the effects of Ipra on inflammation, fibrosis, and cell death in PRAT. HFD-fed mice showed substantial expression of genes related to inflammation (*Emr1*, *Tnf*, *Ccl2*, and *Ccr2*) and fibrosis (*Col1a2*, *Col6a3*, and *Tgfb1*) in PRAT compared to SD-fed mice, with Ipra treatment significantly weakening or tending to weaken the upregulation (Figure 2a and Appendix A). Among the proinflammatory genes, the expression levels of *Cd206*, *Il6*, and *Il1b* were not significantly suppressed by Ipra (Appendix A). HFD-fed mice had decreased gene expression of *Adipoq*, the downregulation of which was weakened by Ipra administration (Figure 2a). Consistent with these results, F4/80 staining of PRAT tissue, a marker of macrophage infiltration, showed that Ipra treatment suppressed macrophage infiltration and crown-like structure (CLS) formation in PRAT (Figure 2b). Sirius red staining revealed that Ipra-treated mice had significantly lower fibrosis in PRAT than HFD-fed mice without Ipra treatment (Figure 2b). Furthermore, Ipra significantly suppressed the number of TUNEL-positive cells in the PRAT of Ipra-treated mice (Figure 2b), consistent with the decrease in the number of CLSs. The expression of high mobility group box protein-1 (HMGB-1), which is secreted by injured and dead cells to enhance inflammatory processes [19], was significantly decreased in the PRAT of Ipra-treated mice (Appendix A).

Flow cytometric analysis performed to examine changes in the properties of adipose tissue macrophages (ATMs) showed that Ipra significantly suppressed the M1/M2 ratio, the ratio of M1-type ATMs (CD11c^+^CD206^-^ cells) to M2-type ATMs (CD11c^-^CD206^+^ cells) in PRAT (Figure 2c). Taken together, Ipra suppressed inflammation, fibrosis, and cell death in PRAT, associated with anti-inflammatory properties of ATMs in PRAT.

### 2.3. Ipra Inhibited Macrophage Infiltration and Proliferative Signals in the Kidney

The current investigated the effects of Ipra administration on inflammation, fibrosis, and proliferative signals in the kidney, associated with DN [20]. PAS staining showed that HFD-fed mice had larger glomeruli than SD-fed mice, suppressed by Ipra treatment (Figure 3a). F4/80 staining to evaluate macrophage infiltration in the kidney and sirius red staining to evaluate fibrosis in the kidney showed that HFD-fed mice had a larger positive staining area in the renal cortex than SD-fed mice, which was suppressed by Ipra treatment (Figure 3a). Gene expression analysis of the kidneys showed that HFD feeding significantly increased the expression levels of proinflammatory genes (*Emr1*, *Tnf*, and *Ccl2*), which was significantly or modestly decreased by Ipra treatment (Figure 3b and Appendix A). HFD feeding significantly increased fatty acid transport-related genes (*Fabp1*) and fatty acid synthesis-related genes (*Fas*) in the kidney, which were suppressed by Ipra treatment, and suppressed lipolysis-related genes (*Cpt1a*), which were increased by Ipra treatment (Appendix A). HFD feeding significantly increased the number of lipid droplets in the renal tubules assessed through PAS staining, which was suppressed by Ipra administration (Appendix A).

We investigated proliferative signaling in the kidney because of a study that suggested that proliferation of GECs precedes glomerulosclerosis and is involved in the pathogenesis of DN [20]. Proliferating cell nuclear antigen (PCNA) expression and p38 mitogen-activated protein kinase (p38 MAPK) phosphorylation were activated in the kidneys of HFD-fed mice but suppressed by Ipra administration (Figure 3c and Appendix A). Immunostaining revealed that the area stained with PCNA antibodies was enlarged in the kidneys, including the glomerular area of HFD-fed mice, which was reversed by Ipra treatment (Figure 3d and Appendix A).

These findings suggest that Ipra treatment suppresses macrophage infiltration and chronic inflammation in the kidney, probably accompanied by suppressing proliferative signals in glomerular cells.

### 2.4. Reduced Leptin Gene Expression in PRAT and Leptin Concentrations in Renal Veins in Ipra-Treated Mice

To determine the mechanism by which Ipra treatment suppressed kidney inflammation and proliferative signaling, we focused on genes related to stimulators of cell proliferation in PRAT. Gene expression levels of *Lep*, *Angptl2*, and *Nampt* in PRAT were significantly increased in HFD-fed mice but were significantly suppressed by Ipra treatment (Figure 4a). Effects of Ipra on *Emr1* and *Lep* gene expression levels in PRAT were still observed even after adjustment for median adipocyte size (Appendix A). After measuring leptin concentrations in the right renal vein to estimate the total amount of leptin circulating in the kidney, we found that HFD-fed mice had significantly higher leptin concentrations than SD-fed mice, which were significantly and suppressed by Ipra treatment (Figure 4b). Although similar changes of leptin concentrations were observed also in systemic circulation assessed in blood obtained from right ventricle, suppression of leptin concentration by Ipra did not reach statistical significance (Figure 4c). Correlation analysis showed a significant positive correlation between renal venous leptin concentrations and UAE (Figure 4d), which was stronger than correlation between leptin concentration in systemic circulation and UAE (Figure 4e).

Unlike other leptin receptor isoforms, ObRa is expressed in the kidney, which differs from ObRb mainly expressed in the hypothalamus [21,22]. ObRa expression was significantly upregulated in kidney of HFD-fed mice and significantly downregulated in Ipra-treated mice, although its expression did not differ significantly within each group (Figure 4f). HFD significantly increased gene expression of *Sl**c5a2* encoding SGLT2 in the kidney, which was suppressed by Ipra treatment (Figure 4f).

These findings suggest that Ipra treatment was likely to suppress leptin action in kidney, at least partly through suppressed expression of leptin in PRAT and leptin receptors in the kidney, contributing to decreased UAE in HFD-fed mice.

### 2.5. Ipra Suppressed Leptin Secretion from PRAT and Proliferative Signaling in Mouse GECs

Finally, we determined whether Ipra treatment suppressed proliferative signaling in mouse glomerular endothelial cells (GECs) through modulating leptin secretion from PRAT, by ex vivo experiments. Leptin concentrations in the conditioned media (CM) prepared from PRAT (PRAT-CM) were higher in HFD-fed mice than in SD-fed mice, and these increases were suppressed by Ipra treatment (Figure 5a). We performed correlation analysis between leptin concentration in PRAT-CM and serum leptin concentration in Ipra-treated mice: Only serum leptin concentration in renal vein was significantly correlated with leptin concentration in PRAT-CM (Appendix A). As previously reported [23], mouse recombinant leptin (mLep) upregulated PCNA expression through p38 MAPK in GECs (Figure 5b). Additionally, the p38 inhibitor SB203580, the PI3K inhibitor LY294002, and the competitive inhibitor of leptin receptor (leptin triple antagonist, LeptA) suppressed leptin-induced p38 MAPK phosphorylation and PCNA expression in GECs (Figure 5c).

P38 phosphorylation and PCNA expression were significantly greater in GECs treated with PRAT-CM from HFD-fed mice than from SD-fed mice, and this upregulation was suppressed by Ipra treatment (Figure 5d and Appendix A–e). Furthermore, SB203580, LY294002, and LeptA blunted the induction of p38 phosphorylation and PCNA expression induced by PRAT-CM from HFD-fed mice (Figure 5d and Appendix A). These findings suggest that leptin from PRAT induces proliferative signals in GECs via p38MAPK, and that Ipra treatment suppresses leptin production from PRAT to inhibit CM-induced up regulation of proliferative signals in GECs.

## 3. Discussion

In the present study, we found that Ipra treatment ameliorated DN in HFD-fed wild-type mice, accompanied by an expansion of PRAT. The expanded PRAT displayed improved insulin signaling and suppressed inflammation and fibrosis, consistent with healthy adipose expansion. PRAT-CM of HFD-fed mice treated with Ipra promoted lower cell proliferation signals in GECs than PRAT-CM from HFD-fed mice without Ipra treatment. These results provide new insights into the mechanism by which SGLT2 inhibitors suppress DN and the establishment of therapies focused on PRAT.

Several reports have shown that PRAT affects the progression of DN. One report found that the improvement in PRAT inflammation through genetic and exogenous PAI-1 inhibition suppressed the development of DN in obese mice [13]. The other report showed that an angiotensin II receptor blocker telmisartan inhibited leptin secretion from PRAT to ameliorate DN, accompanied with suppression of proliferative signaling in the kidney [14]. Given the location of PRAT between the renal fascia and renal capsule, significant amounts of PRAT might activate the renin–angiotensin system in hypertensive and obese rabbits by applying physical pressure to the surrounding blood vessels, exacerbating obesity-related metabolic disorders, such as hypertension, insulin resistance, and atherosclerosis [12]. However, the present study found that DN development was suppressed despite the increase in PRAT amount, suggesting that Ipra-induced functional alternation of PRAT may also affect DN development.

SGLT2 inhibitors as in the present study, thiazolidinediones [24], DPP4 inhibitors [25], and GLP-1 agonists [26] reportedly attenuate albuminuria in obese or diabetic model mice. We proposed that the SGLT2 inhibitor-specific mechanism by which Ipra suppresses DN development in HFD-fed mice involves decreased leptin production from PRAT by Ipra. Leptin has a wide range of effects across organs other than the kidney and regulates the balance between appetite and caloric intake. In the kidney, reports have shown that short-term leptin stimulation activates proliferative signals in GECs by increasing TGF-beta1 expression [23]. The initial proliferation of GECs precedes glomerulosclerosis and has been reported to be among the essential mechanisms for obesity- and diabetes-related nephropathy [20]. In glomerular endothelial cells of diabetic mice, leptin stimulates cellular proliferation, transforming growth factor-beta1 synthesis, and type 4 collagen production, which consequently accelerate glomerulosclerosis [20]. In humans, glomerulosclerosis reportedly correlates with progression of UAE in patients with DN [27]. Then, suppression of leptin-induced activation of proliferative signals in glomerular endothelial cells, as shown in the present study, can lead to reduction of UAE. Other studies also reported that long-term leptin stimulation increased the expression of type 1 and type 4 collagen also in mesangial cells, as well as urinary protein excretion in a blood pressure-independent manner due to glomerulosclerosis progression [28].

Given that leptin concentrations in the renal vein showed solid and positive correlations with UAE, leptin circulating in the kidney may play a crucial role in developing DN in the present study. Because the correlation was weaker in systemic leptin concentration than in the renal vein, reduced production of PRAT-derived leptin by Ipra was more likely to contribute to attenuation of DN in HFD-fed mice than leptin in the systemic circulation. In vitro studies using PRAT-CM and GECs are consistent with this proposed mechanism: PRAT-CM from HFD-fed mice enhanced cell proliferation signaling in GECs via p38MAPK- and leptin-dependent pathways.

As well as leptin, PRAT synthesizes and secretes various adipokines and pro-inflammatory cytokines, including adiponectin, visfatin, resistin, tumor necrosis factor (TNF)-α, interleukin (IL)-6, and -1β, some of whose gene expression in PRAT were altered by Ipra [29]. These cytokines enter nearby kidneys and serve to regulate renal function through endocrine or paracrine pathways [30]. For instance, inhibiting the levels of inflammatory cytokines including IL-6, IL-1b, and TNF-α in PRAT through upregulation of heme oxygenase system reduced renal inflammation and ameliorated diabetic nephropathy in rats [31]. Thus, not only leptin, but also other PRAT-derived adipokines and pro-inflammatory cytokines altered by Ipra could play an inhibitory role in the development of DN of the present study. Precise mechanism by which each PRAT-derived adipokine and pro-inflammatory cytokine affects DN still warrants further studies.

The Ipra-induced functional alternation of PRAT was consistent with “healthy adipose expansion”, an increase in the accumulation without inflammation or fibrosis as we previously reported in Epi and PVAT [15,16,17,18]. These studies suggest that SGLT2 inhibitors increase fat storage capacity in adipocytes, decrease adipocyte death, and suppress macrophage infiltration. Similarly, we observed this healthy adipose expansion in the PRAT of Ipra-treated mice. We previously confirmed that the gene of SGLT2 (*Slc5a2*) was hardly expressed in adipose tissue of mice [16]. Therefore, we consider that Ipra is unlikely to directly inhibit SGLT2 in adipocytes to induce the healthy adipose expansion.

Studies using genetically modified mice suggested the presence of three possible pathologies for this healthy adipose expansion: (i) activation of insulin signaling in adipocytes (observed in adipocytes in adipocyte-specific phosphatase and tensin homolog (PTEN) knockout mice [32]); (ii) reduced macrophage infiltration into adipose tissues (observed in macrophage-inducible C-type lectin (mincle) knockout mice [33]); and (iii) reduced oxidative stress in adipocytes (observed in adipocyte-specific reactive oxygen species removed mice [34]).

As with Epi of PTEN knockout mice, Ipra attenuated insulin resistance of PRAT in HFD-fed mice. Moreover, as with mincle knockout mice, Ipra-treated mice exhibited reduced ATMs in PRAT, indicating amelioration of PRAT inflammation. Although the current study did not examine oxidative stress in PRAT, reports have suggested that induction of 3-HBA suppresses oxidative stress by inhibiting histone deacetylases [35]. The increase we observed in serum 3-HBA levels following Ipra administration may have inhibited oxidative stress in PRAT. Then, the Ipra-induced PRAT expansion was consistent with the pathologies of healthy adipose expansion.

Moreover, we recently reported a novel mechanism by which elevated 3-HBA in Ipra-treated mice contributes to the healthy adipose expansion of Epi by suppressing IL-15 expression in ATMs and increasing the expression of lipogenesis-related genes, such as *Pck1* [15]. Similarly, we observed an increase in serum 3-HBA, suppression of IL-15 expression in PRAT, and an increase in *Pck1*, suggesting that this pathway may have been involved in the pathogenesis of the healthy adipose expansion. Taken together, these various factors, including increased insulin signaling, decreased macrophage infiltration, and increased ketone bodies could contribute to induce the healthy adipose expansion in PRAT of Ipra-treated mice.

Because Ipra treatment caused an elevation in fasting serum NEFA levels in HFD-fed mice via attenuation of hyperinsulinemia, Ipra appeared to enhance lipolysis in adipose tissue. In fact, increase of perilipin 1 in Ipra-treated mice, which allows a low level of basal lipolysis by reducing the access of cytosolic lipases to stored triacylglycerol in lipid droplets in adipocytes [36,37], was consistent with the enhancement of lipolysis. However, the possible increase of lipolysis has a small net effect on adiposity in Ipra-treated mice; increased lipid-storage capacity in adipocytes by improvement of insulin sensitivity may exceed the enhanced lipolysis, resulting in an increase in adiposity in Ipra-treated mice.

We also observed a decrease in TUNEL-positive cells apart from adipose expansion in the PRAT of Ipra-treated mice. These results are consistent with those in our previous reports on Epi [17] and PVAT [18] in Ipra-treated mice, which suggested that Ipra increases the storage capacity of adipocytes and suppresses adipocyte cell death. The PRAT of Ipra-treated mice had markedly reduced expression of HMGB-1, which is secreted by injured and dead cells and enhances inflammatory processes [19]. HMGB-1 is both a hallmark of cell death and a cytokine. Moreover, HMGB1 is one of the endogenous ligands for TLR2, TLR4, and RAGE, while extracellular HMGB-1 binds to target receptors such as TLRs and RAGE, promoting nuclear translocation of transcription factors such as NF-kB [38] and is involved in the pathogenesis of diabetes [39] and renal ischemia–reperfusion injury [40]. Regarding DN, in vitro experiments using tubular epithelial cells and podocytes reported that hyperglycemic stimulation promotes HMGB-1 secretion, promoting increased expression of TLR2 and TLR4, activation of NF-kB, and increased levels of various proinflammatory cytokines [41,42,43]. In vivo experiments showed that administration of HMGB-1 inhibitors improved diabetes-related nephropathy by suppressing TLR activation [44]. Although a detailed pathway could not be established, these findings suggest that Ipra-induced suppression of cell death in PRAT and other gene expression changes in PRAT may have also suppressed the development of DN.

Each observation in the present study was consistent with a conclusion that Ipra reduced urinary albumin excretion as a result of an inter-organ network, which was primally triggered by increased urinary glucose excretion. In particular, we assumed that PRAT mediated the inter-organ communication through secretory factors including leptin and inflammatory cytokines. However, strictly speaking, we could not precisely demonstrate the inter-organ network because interventional in vivo experiments were lacking in the present study. Further experiments are awaited to demonstrate the SGLT2 inhibitor-triggered inter-organ network.

In conclusion, we propose a novel mechanism by which the SGLT2 inhibitor Ipra induces adipocyte hypertrophy without increasing inflammation, fibrosis, or adipocyte death in PRAT of HFD-fed mice. Our findings also imply a novel inter-organ network between kidney and PRAT, possibly mediated by leptin, for the development of DN, and thus suggests new clinical implications for the prevention and treatment of DN through pharmacological intervention.

## 4. Materials and Methods

Additional details of the Materials and Methods are included in the Appendix A.

### 4.1. Animals and Experimental Protocol

Male C57BL/6J WT mice were purchased from CLEA Japan, Inc. (Tokyo, Japan) and Charles River Laboratories Japan, Inc. (Kanagawa, Japan). The animals were allowed free access to water and a standard diet (SD, CE-2; 343 kcal/100 g, 12.6% energy as fat; CLEA Japan, Inc.). Ipra, provided by Astellas Pharma Inc. (Tokyo, Japan), was dissolved in dimethyl sulfoxide (DMSO; Nacalai Tesque, Inc., Kyoto, Japan) at 0.04% (*v*/*v*) and added to the drinking water. During HFD feeding experiments, 8-week-old WT mice were fed an HFD (D12492; 524 kcal/100 g, 60% energy as fat; Research Diets, Inc., New Brunswick, NJ, USA) for 14 weeks, followed by an HFD with the vehicle or Ipra for 6 weeks. Ipra concentrations in the drinking water were changed every week based on daily water consumption and body weight, adjusting to 10 mg/kg/day. Age-matched control male WT mice were fed an SD throughout the experiment period. Body weight and blood glucose were measured every 2 weeks. At the end of the experiment, animals were sacrificed using intraperitoneal pentobarbital anesthesia (30 mg/kg) after 16 h of fasting. Both kidneys, PRAT, Epi, and liver were carefully dissected and removed. After careful removal of the vessels and connective tissue surrounding adipose tissue, the remaining PRAT, Epi, and liver were weighed on an electronic balance. The kidney and PRAT were divided for histological and mRNA/protein analyses, respectively.

### 4.2. Materials

The p38 MAPK inhibitor SB 203,580 and PI3K inhibitor LY294002 were obtained from Cell Signaling Technology (Danvers, MA, USA). The leptin antagonist triple mutant (leptA), a mutant leptin analog that functions as a competitive inhibitor, was purchased from PROSPEC (Rehovot, Israel). Recombinant mouse leptin was purchased from PeproTech (Rocky Hill, NJ, USA).

### 4.3. Statistical Analysis

Normally distributed values were expressed as mean ± standard error of the mean and compared using Student’s t-test or analysis of variance with post hoc testing. Non-Gaussian distributed values were expressed as box-and-whisker plots with median values and 10th, 25th, 75th, and 90th percentiles. Nonparametric statistical analysis was performed using the Mann–Whitney or Kruskal–Wallis tests with Dunn post hoc tests. All statistical analyses were performed using Prism 7 (GraphPad Software, Inc., San Diego, CA, USA), with *p* < 0.05 indicating statistical significance.

## Figures and Tables

**Figure 1 ijms-22-07329-f001:**
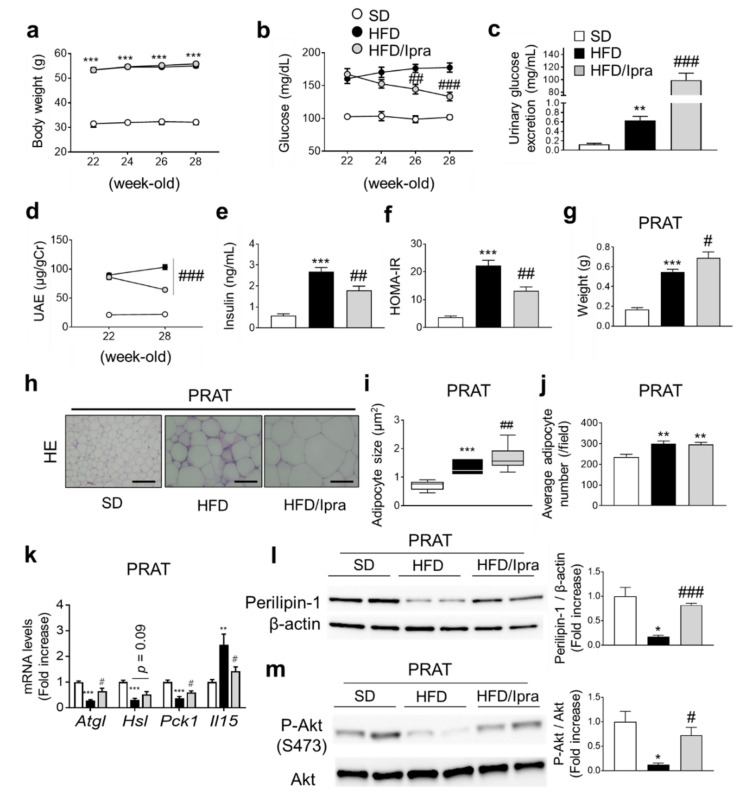
Ipra ameliorates HFD-induced urinary albumin excretion associated with increased adipocyte size and enhanced insulin signaling in PRAT. Changes in (**a**) body weight and (**b**) blood glucose in SD- or HFD-fed WT mice during treatment with or without Ipra from 22 to 28 weeks of age. (**c**) Urinary glucose excretion after 3 weeks of Ipra treatment. (**d**) Changes in urinary albumin excretion (UAE) before and after Ipra treatment for 6 weeks. (**e**) Fasting plasma insulin concentration, (**f**) HOMA-IR, and (**g**) PRAT weight after 6 weeks of Ipra treatment. (**h**) Hematoxylin and eosin staining and quantification of (**i**) adipocyte size and (**j**) number of adipocytes per field in PRAT. (**k**) Expression levels of lipolysis- and lipogenesis-related genes in PRAT of SD- or HFD-fed WT mice after 6 weeks of Ipra treatment. Representative blots and quantitative data of (**l**) perilipin-1 and (**m**) insulin-induced Akt phosphorylation (p-Akt) in PRAT. WT, wild-type; SD, standard diet; HFD, high-fat diet; Ipra, ipragliflozin; PRAT, perirenal adipose tissue; HOMA-IR, homeostatic model assessment for insulin resistance. Scale bar, 1 µm. Original magnification, ×200. * *p* < 0.05, ** *p* < 0.01, *** *p* < 0.001 vs. SD. # *p* < 0.05, ## *p* < 0.01, ### *p* < 0.001 vs. HFD. *n* = 6 (**a**–**d**,**g**,**k**) *n* = 9–11 (**e**,**f**,**i**,**j**).

**Figure 2 ijms-22-07329-f002:**
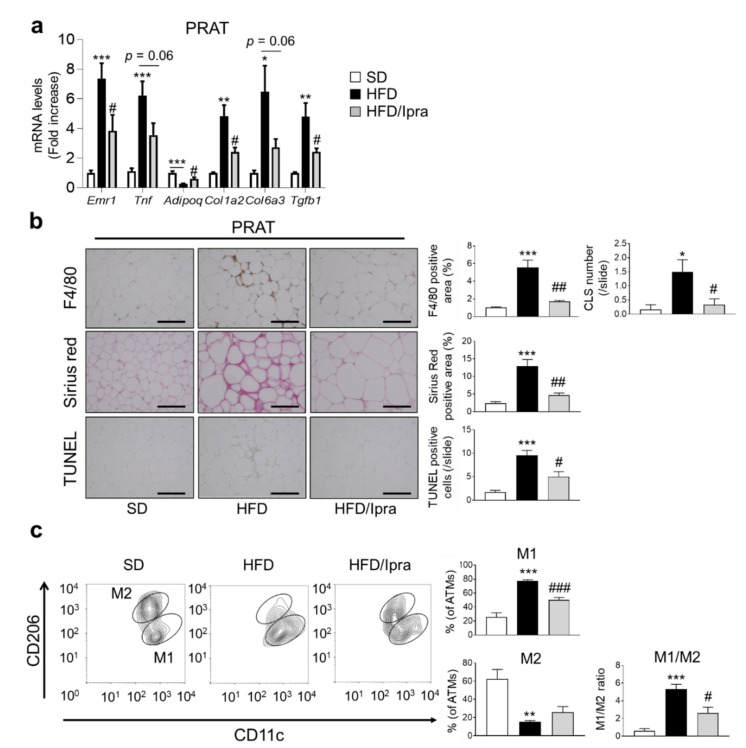
Ipra attenuates HFD-induced inflammation, fibrosis, and cell death in PRAT. (**a**) Expression levels of inflammation- and fibrosis-related genes in PRAT of SD- or HFD-fed WT mice after 6 weeks of Ipra treatment. (**b**) Representative images and quantitative data of F4/80 immunostaining, Sirius red staining, and TUNEL staining in PRAT. (**c**) Representative plots and quantification of flow cytometry for M1-like (CD11c^+^CD206^−^ cells) and M2-like (CD11c^−^CD206^+^ cells) ATMs (CD45^+^CD11b^+^F4/80^+^ cells) in PRAT and M1-like/M2-like ratio. WT, wild-type; SD, standard diet; HFD, high-fat diet; Ipra, ipragliflozin; PRAT, perirenal adipose tissue; TUNEL, TdT-mediated dUTP nick end labeling; ATMs, adipose tissue macrophages. Original magnification, ×200. Scale bar, 1 µm. * *p* < 0.05, ** *p* < 0.01, *** *p* < 0.001 vs. SD. # *p* < 0.05, ## *p* < 0.01, ### *p* < 0.001 vs. HFD. *n* = 6.

**Figure 3 ijms-22-07329-f003:**
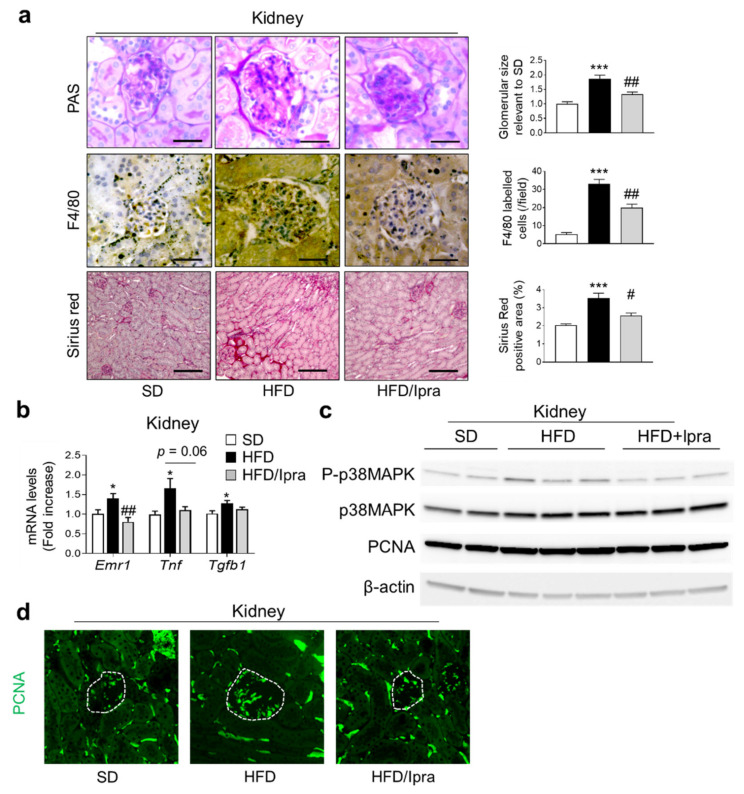
Ipra inhibits macrophage infiltration and proliferative signals in the kidney. (**a**) Representative images and quantification of PAS staining, F4/80 immunostaining, and sirius red staining in kidney. (**b**) Expression levels of inflammation- and fibrosis-related genes in the kidney of SD- or HFD-fed WT mice after 6 weeks of Ipra treatment. (**c**) Representative blots of phosphorylated/total p38 mitogen-activated protein kinase (p38 MAPK) and PCNA in the kidney. (**d**) Representative pictures of fluorescent immunostaining of PCNA (green). Inside of dotted lines indicate the glomerular tuft areas. WT, wild-type; SD, standard diet; HFD, high-fat diet; Ipra, ipragliflozin; PRAT, perirenal adipose tissue. Scale bar, 50 µm (PAS and F4/80), 250 µm (Sirius red). * *p* < 0.05, *** *p* < 0.001 vs. SD. # *p* < 0.05, ## *p* < 0.01 vs. HFD. *n* = 6.

**Figure 4 ijms-22-07329-f004:**
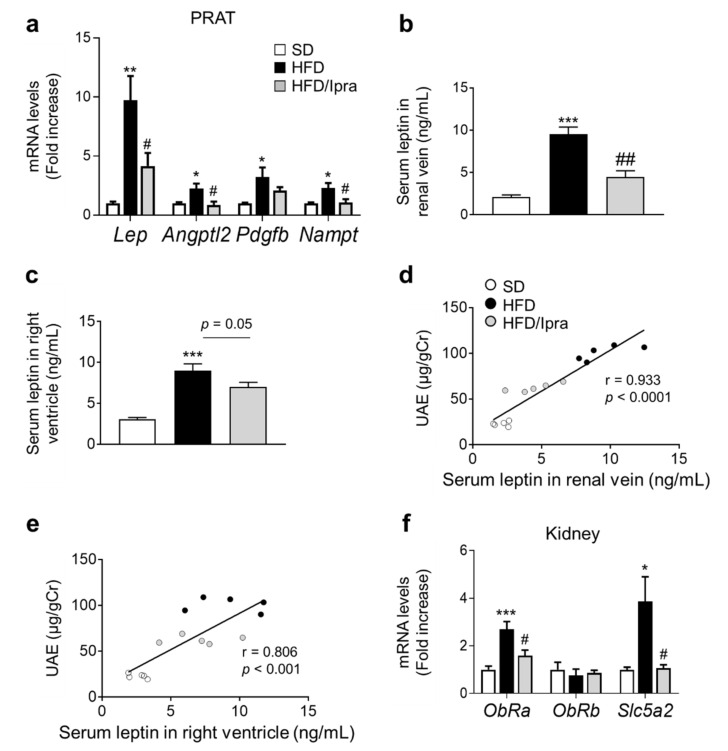
Leptin gene expression in PRAT and leptin levels in renal veins are reduced in Ipra-treated mice. (**a**) Expression levels of genes related to proliferative signaling in PRAT SD- or HFD-fed WT mice after 6 weeks of Ipra treatment. Serum leptin levels in (**b**) right ventricle and (**c**) right renal vein in SD- or HFD-fed WT mice during Ipra treatment for 6 weeks. (**d**) Correlation analysis between renal venous leptin concentration and urinary albumin excretion (UAE). (**e**) Correlation analysis between serum leptin concentration in right ventricle and UAE. (**f**) Gene expression levels of *ObRa*, *ObRb*, and *Sl**c5a2*. WT, wild-type; SD, standard diet; HFD, high-fat diet; Ipra, ipragliflozin; PRAT, perirenal adipose tissue. * *p* < 0.05, ** *p* < 0.01, *** *p* < 0.001 vs. SD. # *p* < 0.05, ## *p* < 0.01 vs. HFD. *n* = 6 (**a**,**b**,**e**) *n* = 5 (**c**).

**Figure 5 ijms-22-07329-f005:**
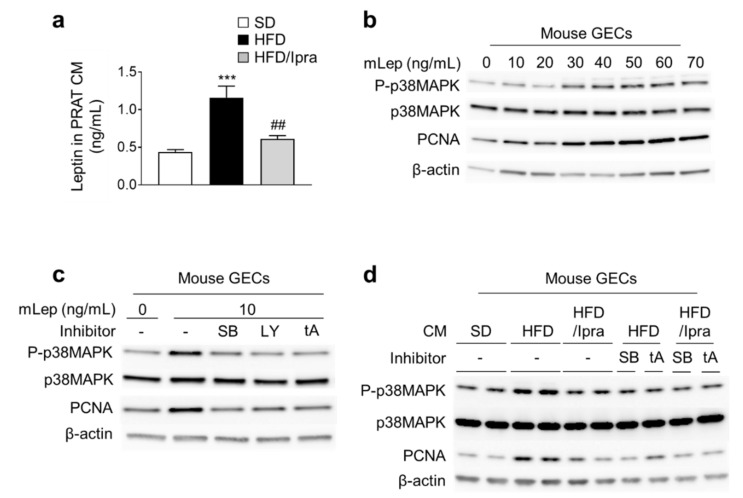
Ipra suppresses leptin secretion from PRAT and proliferative signaling in mouse GECs. (**a**) Leptin concentrations in conditioned media (CM) made from PRAT. (**b**) Western blotting of phosphorylated and total p38 mitogen-activated protein kinase (p38 MAPK), and proliferating cell nuclear antigen (PCNA) in mouse GECs treated with mouse recombinant leptin (mLep). (**c**) Western blotting of phosphorylated and total p38 MAPK, and PCNA in mouse GECs treated with mLep after pre-incubation with SB203580 (SB), LY294002 (LY), and leptin triple antagonist (tA), which functions as a leptin competitive inhibitor. (**d**) Western blotting of phosphorylated and total p38 MAPK, and PCNA in mouse GECs treated with PRAT-CM after pre-incubation with SB and tA. *** *p* < 0.001 vs. SD. ## *p* < 0.01 vs. HFD. *n* = 6 (**a**).

## Data Availability

The data that support the findings of this study are available from the corresponding author upon reasonable request.

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
