# Peer review of "Ipragliflozin Ameliorates Diabetic Nephropathy Associated with Perirenal Adipose Expansion in Mice"

_ijms, 2021, doi:10.3390/ijms22147329_

Round 1
Reviewer 1 Report
In this paper Okuma et all present data on changes in perirenal fat (PRAT) in high-fat diet (HFD)-fed mice induced by treatment with an SGLT2 inhibitor. They find that adipocyte size was increased, and inflammation, fibrosis, and adipocyte death were suppressed in treated animals and that in the conditioned medium concentration of leptin was significantly lower than in non- treated animals. Interestingly serum leptin concentration in renal vein correlated with UAE.
Some comments on the paper:
Background - it is important to clearly state if the findings presented are from animal models or from humans, for example, to my knowledge, there has not been any ectopic fat accumulation found in the human diabetic kidney. Please clarify.
With regards to mechanisms the SGLT2-inhibitors have multiple effects and one important mechanism that should be highlighted in this context is the effect on tubuloglomerular feedback and hyperfiltration which is closely connected to potential effects on hyperfiltration and proteinuria Vallon V, Thomson SC. The tubular hypothesis of nephron filtration and diabetic kidney disease. Nat Rev Nephrol. 2020 Jun;16(6):317-336. Please add.
It would also be of interest for the reader to have some insights on the known effects of SGLT2s on other fat depots in the background. Please add.
The aim and objective of this study is not clearly stated for example at the end of the background section. Please revise this and add what the aims and objectives of the study were and move the section “in the present study, we found that” to the discussions section where it will fit nicely.
Results
The authors show an effect of SGLT2s on adipocyte size in PRAT (increase). What would the results be if mRNA levels were expressed per adipocyte size instead?
In the extensive results section there is data on findings in the kidney (macrophage infiltration and proliferative signaling) and in the renal veins (leptin concentrations) and it is not clear to me how we these findings may effects of the alterations by SGLT2 on the PRAT instead of effects direct effects on the kidney. Please clarify.
Discussion
Please add the summary from the background here.
The first statement in the discussion is that several reports have shown that PRAT affects the progressions of DN. Please add some of these references for completeness.
In the next section the authors state that effects of SGLT2s on PRAT could contribute to the effects of SGLT2s to suppress development of DN (or at least reduce UAE). Effects on leptin are discussed in detail. Please comment specifically on how (mechanism) the effects on leptin (overall and in the kidney) could be related to the effects on urinary albumin excretion.
Several interesting and intriguing findings are described. Please comments if the findings in this paper are results of an inter-organ network or if these might be parallel phenomenon with direct effects on the kidney and on PRAT and why you have come to this conclusion.
Author Response
Background - it is important to clearly state if the findings presented are from animal models or from humans, for example, to my knowledge, there has not been any ectopic fat accumulation found in the human diabetic kidney. Please clarify.
As pointed, there has been no evidence for ectopic fat accumulation in the human diabetic kidney. Then, we revised the sentences to clarify which evidence was obtained from humans or mice.
With regards to mechanisms the SGLT2-inhibitors have multiple effects and one important mechanism that should be highlighted in this context is the effect on tubuloglomerular feedback and hyperfiltration which is closely connected to potential effects on hyperfiltration and proteinuria Vallon V, Thomson SC. The tubular hypothesis of nephron filtration and diabetic kidney disease. Nat Rev Nephrol. 2020 Jun;16(6):317-336. Please add.
We have added the suggested reference in Introduction part.
It would also be of interest for the reader to have some insights on the known effects of SGLT2s on other fat depots in the background. Please add.
SGLT2 inhibitors reportedly enhanced appetite and promoted epididymal fat (Epi) accumulation without deteriorating inflammation or fibrosis in mice [1-3], referred to as “healthy adipose expansion.” In terms of interactions between adipose tissue with other organs, we further showed that Ipra gave rise to healthy adipose expansion in mice perivascular adipose tissue while implanting adipose tissue with healthy adipose expansion into the perivascular area of an atherosclerosis model suppresses vascular remodeling [4]. We also reported that Ipra increased fat accumulation in Epi, but decreased fat accumulation in liver [3]. Thus, it has been suggested that changes in the quality and quantity of fat within or surrounding each organ may underlie the mechanism of organ protection by SGLT2 inhibitors. We incorporated the description into Introduction part.
The aim and objective of this study is not clearly stated for example at the end of the background section. Please revise this and add what the aims and objectives of the study were and move the section “in the present study, we found that” to the discussions section where it will fit nicely.
The aim and objective of this study were added at the end of the Introduction section. The paragraph summarizing the results of this study, which used to be at the end of Introduction, has been moved to the beginning of the Discussion part.
Results
The authors show an effect of SGLT2s on adipocyte size in PRAT (increase). What would the results be if mRNA levels were expressed per adipocyte size instead?
As shown in Fig. 1i, the median adipocyte size of PRAT in Ipra-treated mice was approximately 1.23-fold larger than that in non-Ipra-treated mice. Effects of Ipra on Emr1 and Lep expression levels in PRAT were still observed even after adjustment for median adipocyte size. These results have been added to new Supplementary Fig. 4c.
In the extensive results section there is data on findings in the kidney (macrophage infiltration and proliferative signaling) and in the renal veins (leptin concentrations) and it is not clear to me how we these findings may effects of the alterations by SGLT2 on the PRAT instead of effects direct effects on the kidney. Please clarify.
As shown in Fig. 4b and c, Ipra treatment reduced serum leptin concentration in renal vein more than in right ventricle. An additional analysis showed that serum leptin concentration in renal vein was significantly correlated with leptin concentration in conditioned medium from PRAT, but not with serum leptin concentration in right ventricle (new Supplementary Fig. 4d). It suggests that major source of leptin in renal vein is PRAT. Given that urinary albumin excretion (UAE) showed tight positive correlation with serum leptin concentration in renal vein, reduction of PRAT-derived leptin production by Ipra was likely to play an important role in suppression of urinary albumin excretion. However, we cannot exclude the possibility of other direct effects on the kidney: for example, Ipra may affect tubuloglomerular feedback to attenuate hyperfiltration and proteinuria as previously reported [5]. We modified the sentence describing PRAT-derived leptin’s significance in Discussion part.
Discussion
Please add the summary from the background here.
The paragraph summarizing the results of this study, which used to be at the end of Introduction, has been moved to the beginning of the Discussion part.
The first statement in the discussion is that several reports have shown that PRAT affects the progressions of DN. Please add some of these references for completeness.
A previous report [6] showing that telmisartan inhibited DN progression associated with reduced PRAT inflammation was added.
In the next section the authors state that effects of SGLT2s on PRAT could contribute to the effects of SGLT2s to suppress development of DN (or at least reduce UAE). Effects on leptin are discussed in detail. Please comment specifically on how (mechanism) the effects on leptin (overall and in the kidney) could be related to the effects on urinary albumin excretion.
In glomerular endothelial cells of diabetic mice, leptin stimulates cellular proliferation, transforming growth factor-beta1 synthesis, and type IV collagen production, which consequently accelerate glomerulosclerosis [7]. In humans, glomerulosclerosis reportedly correlates with progression of urinary albumin excretion in patients with DN [8]. Then, suppression of leptin-induced activation of proliferative signals in glomerular endothelial cells, as shown in the present study, can lead to reduction of urinary albumin excretion. We incorporated the statement in Discussion part.
Several interesting and intriguing findings are described. Please comments if the findings in this paper are results of an inter-organ network or if these might be parallel phenomenon with direct effects on the kidney and on PRAT and why you have come to this conclusion.
Each observation in the present study was consistent with a conclusion that Ipra reduced urinary albumin excretion as a result of an inter-organ network, which was primally triggered by increased urinary glucose excretion. Especially, we assumed that PRAT mediated the inter-organ communication through secretory factors including leptin and inflammatory cytokines. However, strictly speaking, we could not precisely demonstrate the inter-organ network by intervention experiments in vivo. Then, further experiments are awaited to demonstrate the SGLT2 inhibitors-triggered inter-organ network. We incorporated these sentences as study limitations into Discussion part.
REFERENCES
- Miyachi, Y.; Tsuchiya, K.; Shiba, K.; Mori, K.; Komiya, C.; Ogasawara, N.; Ogawa, Y., A reduced M1-like/M2-like ratio of macrophages in healthy adipose tissue expansion during SGLT2 inhibition. Scientific reports 2018, 8, (1), 16113.
- Shiba, K.; Tsuchiya, K.; Komiya, C.; Miyachi, Y.; Mori, K.; Shimazu, N.; Yamaguchi, S.; Ogasawara, N.; Katoh, M.; Itoh, M.; Suganami, T.; Ogawa, Y., Canagliflozin, an SGLT2 inhibitor, attenuates the development of hepatocellular carcinoma in a mouse model of human NASH. Scientific reports 2018, 8, (1), 2362.
- Komiya, C.; Tsuchiya, K.; Shiba, K.; Miyachi, Y.; Furuke, S.; Shimazu, N.; Yamaguchi, S.; Kanno, K.; Ogawa, Y., Ipragliflozin Improves Hepatic Steatosis in Obese Mice and Liver Dysfunction in Type 2 Diabetic Patients Irrespective of Body Weight Reduction. PLoS One 2016, 11, (3), e0151511.
- Mori, K.; Tsuchiya, K.; Nakamura, S.; Miyachi, Y.; Shiba, K.; Ogawa, Y.; Kitamura, K., Ipragliflozin-induced adipose expansion inhibits cuff-induced vascular remodeling in mice. Cardiovasc Diabetol 2019, 18, (1), 83.
- Vallon, V.; Thomson, S. C., The tubular hypothesis of nephron filtration and diabetic kidney disease. Nat Rev Nephrol 2020, 16, (6), 317-336.
- Li, H.; Li, M.; Liu, P.; Wang, Y.; Zhang, H.; Li, H.; Yang, S.; Song, Y.; Yin, Y.; Gao, L.; Cheng, S.; Cai, J.; Tian, G., Telmisartan Ameliorates Nephropathy in Metabolic Syndrome by Reducing Leptin Release From Perirenal Adipose Tissue. Hypertension 2016, 68, (2), 478-90.
- Johnson, R. J., What mediates progressive glomerulosclerosis? The glomerular endothelium comes of age. Am J Pathol 1997, 151, (5), 1179-81.
- Shimizu, M.; Furuichi, K.; Toyama, T.; Kitajima, S.; Hara, A.; Kitagawa, K.; Iwata, Y.; Sakai, N.; Takamura, T.; Yoshimura, M.; Yokoyama, H.; Kaneko, S.; Wada, T.; Kanazawa Study Group for Renal, D.; Hypertension, Long-term outcomes of Japanese type 2 diabetic patients with biopsy-proven diabetic nephropathy. Diabetes Care 2013, 36, (11), 3655-62.
Reviewer 2 Report
This is an experimental study examining the effect of an SGLT2 inhibitor, ipragliflozin, on the perirenal fat in high fat diet-induced diabetic mouse. Ipragliflozin reduced blood glucose but not body weight. Urinary albumin excretion reduced after ipragliflozin and glomerular hypertrophy was ameliorated. Ipragliflozin increased the adipocyte size in perirenal fat and suppressed transcripts related to inflammation and fibrosis. Also, they demonstrated that leptin concentration was significantly related to urinary albumin, and that leptin concentration in conditioned medium from perirenal fat from Ipra-treated mice was significantly lower than that from non-treated DN mice. Finally, they nicely showed that p38MAPK and PCNA is involved in the leptin-induced proliferative signals in glomerular endothelial cells. Several comments from the reviewer are listed below;
- The author described that Ipra increased lipolysis (Fig. S1a). However, the PRAT weight significantly increased together with the induction of perilipin-1 after Ipra. In addition, HFD did not induce fat accumulation in PRAT as shown in perilipin-1. Please discuss this discrepancy of lipid metabolism in PRAT.
- The author suggested that Ipra suppressed leptin action through the downregulation in PRAT (pages 7 to 8). Since leptin is expressed in white adipose tissue throughout the body, the effect of Ipra on adipose tissue might not be specific to PRAT. Considering the anatomical distribution of the renal arteries and veins (interlobular artery – efferent – glomerulus – afferent - interlobular vein – finally to the renal vein), the reviewer wonder whether the leptin concentration in the renal vain truly reflect the leptin action in the kidney.
- Please discuss whether the induction of “healthy adipocyte” of Ipra is the direct effect on PRAT or indirect effect (i.e. lipid metabolism or glucose metabolism)
Author Response
The author described that Ipra increased lipolysis (Fig. S1a). However, the PRAT weight significantly increased together with the induction of perilipin-1 after Ipra. In addition, HFD did not induce fat accumulation in PRAT as shown in perilipin-1. Please discuss this discrepancy of lipid metabolism in PRAT.
Each observation in the present study was consistent with a conclusion that Ipra re-duced urinary albumin excretion as a result of an inter-organ network, which was primally triggered by increased urinary glucose excretion. Especially, we assumed that PRAT mediated the inter-organ communication through secretory factors including leptin and inflammatory cytokines. However, strictly speaking, we could not precisely demon-strate the inter-organ network because interventional in vivo experiments were lacking in the present study. Further experiments are awaited to demonstrate the SGLT2 inhibi-tors-triggered inter-organ network.
We incorporated the description in Discussion part.
The author suggested that Ipra suppressed leptin action through the downregulation in PRAT (pages 7 to 8). Since leptin is expressed in white adipose tissue throughout the body, the effect of Ipra on adipose tissue might not be specific to PRAT. Considering the anatomical distribution of the renal arteries and veins (interlobular artery – efferent – glomerulus – afferent - interlobular vein – finally to the renal vein), the reviewer wonder whether the leptin concentration in the renal vain truly reflect the leptin action in the kidney.
As pointed, in order to precisely assess leptin action in the kidney, leptin concentration in renal arteries or the kidney tissue may be more anatomically appropriate. However, we could not assess leptin concentration using these materials due to technical difficulties.
As shown in Fig. 4b and c, Ipra treatment reduced serum leptin concentration in renal veins more than in right ventricle. An additional analysis showed that serum leptin concentration in renal veins was significantly correlated with leptin concentration in conditioned medium from PRAT, but not with serum leptin concentration in right ventricle (new Supplementary Fig. 4d). It suggests that PRAT-derived leptin mainly accounted for leptin in renal vein, at least more than leptin in systemic circulation. Given that urinary albumin excretion showed solid positive correlation with serum leptin concentration in renal veins, reduction of PRAT-derived leptin production by Ipra, which could be assessed by leptin concentration in renal veins, was likely to play an important role in suppression of urinary albumin excretion.
Please discuss whether the induction of “healthy adipocyte” of Ipra is the direct effect on PRAT or indirect effect (i.e. lipid metabolism or glucose metabolism)
We previously confirmed that gene of SGLT2 (Slc5a2) hardly expressed in adipose tissue of mice [1]. Therefore, we consider that Ipra is unlikely to directly inhibit SGLT2 in adipocytes to induce the healthy adipose expansion.
Studies using genetically modified mice suggested the presence of three possible pathologies for this healthy adipose expansion: i) activation of insulin signaling in adipocytes [observed in adipocytes in adipocyte-specific phosphatase and tensin homolog (PTEN) knockout mice [2]]; ii) reduced macrophage infiltration into adipose tissues [observed in macrophage-inducible C-type lectin (mincle) knockout mice [3]]; and iii) reduced oxidative stress in adipocytes [observed in adipocyte-specific reactive oxygen species removed mice [4]].
As with Epi of PTEN knockout mice, Ipra attenuated insulin resistance of PRAT in HFD-fed mice. Moreover, as with mincle knockout mice, Ipra-treated mice exhibited reduced ATMs in PRAT, indicating amelioration of PRAT inflammation. Although the cur-rent study did not examine oxidative stress in PRAT, reports have suggested that induction of 3-HBA suppresses oxidative stress by inhibiting histone deacetylases [5]. The increase we observed in serum 3-HBA levels following Ipra administration may have inhibited oxidative stress in PRAT. Then, the Ipra-induced PRAT expansion was consistent with the pathologies of healthy adipose expansion.
Moreover, we recently reported a novel mechanism by which elevated 3-HBA in Ipra-treated mice contributes to the healthy adipose expansion of Epi by suppressing IL-15 expression in ATMs and increasing the expression of lipogenesis-related genes, such as Pck1 [6]. Similarly, we observed an increase in serum 3-HBA, suppression of IL-15 expression in PRAT, and an increase in Pck1, suggesting that this pathway may have been involved in the pathogenesis of the healthy adipose expansion. Taken together, these various factors, including increased insulin signaling, decreased macro-phage infiltration, and increased ketone bodies could contribute to induce the healthy adipose expansion in PRAT of Ipra-treated mice.
We partly revised description regarding to mechanisms of the healthy adipose expansion in Discussion part.
REFERENCES
- Shiba, K.; Tsuchiya, K.; Komiya, C.; Miyachi, Y.; Mori, K.; Shimazu, N.; Yamaguchi, S.; Ogasawara, N.; Katoh, M.; Itoh, M.; Suganami, T.; Ogawa, Y., Canagliflozin, an SGLT2 inhibitor, attenuates the development of hepatocellular carcinoma in a mouse model of human NASH. Scientific reports 2018, 8, (1), 2362.
- Morley, T. S.; Xia, J. Y.; Scherer, P. E., Selective enhancement of insulin sensitivity in the mature adipocyte is sufficient for systemic metabolic improvements. Nature communications 2015, 6, 7906.
- Tanaka, M.; Ikeda, K.; Suganami, T.; Komiya, C.; Ochi, K.; Shirakawa, I.; Hamaguchi, M.; Nishimura, S.; Manabe, I.; Matsuda, T.; Kimura, K.; Inoue, H.; Inagaki, Y.; Aoe, S.; Yamasaki, S.; Ogawa, Y., Macrophage-inducible C-type lectin underlies obesity-induced adipose tissue fibrosis. Nature communications 2014, 5, 4982.
- Okuno, Y.; Fukuhara, A.; Hashimoto, E.; Kobayashi, H.; Kobayashi, S.; Otsuki, M.; Shimomura, I., Oxidative Stress Inhibits Healthy Adipose Expansion Through Suppression of SREBF1-Mediated Lipogenic Pathway. Diabetes 2018, 67, (6), 1113-1127.
- Shimazu, T.; Hirschey, M. D.; Newman, J.; He, W.; Shirakawa, K.; Le Moan, N.; Grueter, C. A.; Lim, H.; Saunders, L. R.; Stevens, R. D.; Newgard, C. B.; Farese, R. V., Jr.; de Cabo, R.; Ulrich, S.; Akassoglou, K.; Verdin, E., Suppression of oxidative stress by beta-hydroxybutyrate, an endogenous histone deacetylase inhibitor. Science 2013, 339, (6116), 211-4.
- Miyachi, Y.; Tsuchiya, K.; Shiba, K.; Mori, K.; Komiya, C.; Ogasawara, N.; Ogawa, Y., A reduced M1-like/M2-like ratio of macrophages in healthy adipose tissue expansion during SGLT2 inhibition. Scientific reports 2018, 8, (1), 16113.
This manuscript is a resubmission of an earlier submission. The following is a list of the peer review reports and author responses from that submission.